# The emotional mechanism underlying the adverse effect of social exclusion on working memory performance: A tDCS study

Qingguo Ma[1,2,3]*, Yu Pang[1,2]

**1** School of Management, Zhejiang University of Technology, Hangzhou, Zhejiang, China, **2** Institute of Neural Management Sciences, Zhejiang University of Technology, Hangzhou, Zhejiang, China, **3** School of Management, Zhejiang University, Hangzhou, Zhejiang, China

* maqingguo3669@zju.edu.cn

**Data Availability Statement:** All relevant data are within the paper and its Supporting information file.

**Funding:** This study was supported by the National Natural Science Foundation of China (No.71942004). The funders had no role in study

## Abstract

Social exclusion has been found to impair working memory (WM). However, the emotional mechanism underlying this adverse effect remains unclear. Besides, little is known about how to alleviate this adverse effect. In the current study, 128 participants were randomly assigned to a social excluded group or an included group while they received anodal tran-scranial direct current stimulation (tDCS) or sham tDCS over the right ventrolateral pre-frontal cortex (rVLPFC), then they completed the 2-back task. ANOVA results showed that under the sham tDCS condition, mood rating score and 2-back task accuracy of excluded participants were both lower than included participants, and after anodal tDCS, mood rating score and 2-back task accuracy of excluded participants were both higher compared to sham tDCS. Besides, the mediation model showed that negative emotion mediated the relationship between social exclusion and WM under the sham tDCS condition, while the mediating effect disappeared under the anodal tDCS condition. Based on these results, we argued that anodal tDCS over the rVLPFC could alleviate the adverse effect of social exclusion on WM by reducing negative emotion. These findings contributed to further understanding of the emotional mechanism underlying the adverse effect of social exclusion on WM, and providing a clinical treatment in response to social exclusion.

## Introduction

As social beings, humans rely on group life for their health and well-being, and being accepted into a social group is an important goal of human striving [1]. Accordingly, humans have developed an inherent and fundamental need to belong [2]. However, social exclusion, which is an aversive but prevalent phenomenon in daily life, has been proved to thwart the need to belong [3, 4]. What's more, social exclusion may affect not only need-threat but also cognitive functions, particularly executive functions. To date, numerous studies have found the deleterious effects of social exclusion on executive functions, such as response inhibition [5], conflict monitoring [6] and interference control [7, 8]. Recently, some researchers are beginning to

design, data collection and analysis, decision to publish, or preparation of the manuscript.

**Competing interests:** The authors have declared that no competing interests exist.

focus on the impact of social exclusion on working memory, a key component of executive function [9, 10].

Working memory (WM) refers to the structures and processes used to temporarily store and manipulate information in the face of ongoing processing and distraction [11]. WM is widely considered to play a key role in a number of basic cognitive functions, such as language comprehension, learning, reasoning, thinking, action planning and other complex cognitive tasks [12]. Currently, several studies have explored the potential effect of social exclusion on WM. For instance, Buelow et al. [13] used the Digit Span subtest from the Wechsler Adult Intelligence Scale-IV [14] to assess WM, and their results showed that excluded participants had poor Digit Span subtest performance, revealing the adverse effect of social exclusion on WM. In addition, Xu et al. [9] used the Lateralized Change Detection Task to measure the WM, and found that poor WM performance caused by social exclusion was attributed to both reduced storage capacity and impaired attentional filtering ability.

The mechanism underlying the adverse effect of social exclusion on WM performance is mainly understood from the perspective of limited cognitive resources. As a typical negative social experience, social exclusion has been proved to induce negative emotions such as anxiety, sadness and depression [15]. Excluded participants would devote their own regulatory resources to stifle these negative emotions [9]. This emotion regulation process would consume individuals' limited cognitive resources and is thought to compete with WM tasks, leading to the impaired WM performance [16]. However, most studies seem to focus on this resource-consuming process and ignore the effect of negative emotion per se in the adverse effect of social exclusion on WM performance. Several studies have suggested that negative emotion would impair WM performance by disturbing several specific aspects of WM. For instance, Beckwe and Deroost [17] pointed out that the updating capacity of working memory was disrupted after worry induction, and this detrimental effect was irrelevant to individuals' inherent tendency to worry. Figueira et al [18] used contralateral delay activity (CDA) as a neurophysiological index of the representation of the task-relevant items held in working memory, and found that the unpleasant mood was related to reduced working memory capacity (WMC). Taken together, it's worth further exploring the role of negative emotion in the adverse effect of social exclusion on WM performance.

Given the importance of working memory and pervasiveness of social exclusion, it is essential to find a potential solution to alleviate the adverse effect of social exclusion on WM performance. However, few studies have focused on this issue so far, and our study aimed to explore this issue by considering the neural intervention. Neuroimaging studies have indicated that the right ventrolateral prefrontal cortex (rVLPFC) exhibits increased activity in response to social exclusion [19, 20], and the enhanced activation of the rVLPFC was negatively correlated with the self-reported distress [21]. Besides, several transcranial direct current stimulation (tDCS) studies have reported a causal relationship between rVLPFC activity and self-reported negative emotion [15]: anodal stimulation of rVLPFC could reduce negative emotion responses to social exclusion [22], while cathodal stimulation over the rVLPFC could boost negative emotion responses to social exclusion [23]. Recently, researchers suggested that anodal tDCS over the rVLPFC could also reduce the behavioral aggression caused by social exclusion [24, 25]. Accordingly, the present study would like to test whether anodal tDCS over the rVLPFC could also alleviate the adverse effect of social exclusion on WM performance.

In sum, this study aimed to explore the emotional mechanism underlying the adverse effect of social exclusion on WM performance and investigate the potential solution of neural intervention to alleviate this impairment. To this end, we manipulated social exclusion among the participants with the typical Cyberball paradigm and used N-back task to assess their WM performance. Meanwhile, we applied the anodal (vs sham) tDCS over the rVLPFC during the

Cyberball task. Based on the findings from previous studies that (1) social exclusion would induce negative emotion (e.g., anxiety, sadness, depression), and (2) negative emotion would impair WM performance, we hypothesized that negative emotions induced by social exclusion mediate the relationship between social exclusion and WM performance. Besides, according to the findings from previous studies that anodal stimulation over the rVLPFC would reduce the negative emotion responses and behavior aggression elicited by social exclusion, we further hypothesized that anodal stimulation over the rVLPFC would also alleviate the adverse effect of social exclusion on WM performance by reducing negative emotion.

## Methods

### Participants

G*power 3.1 was used to determine the sample size [26]. Under the premise of power $(1-\beta)$ = 0.95, $\alpha$ = 0.05 and effect size f = 0.4 [27], a minimum of 84 participants were needed. Based on this, 128 healthy students from Zhejiang University of Technology were recruited (45% female; $M_{age}$ = 21.8 years; $SD_{age}$ = 2.09 years). They were randomly assigned to either the social inclusion or social exclusion group. These two groups were further randomly divided into the anodal and sham stimulation groups. As a result, all the participants were divided into four groups with 32 in each group. All participants provided written informed consent, and the study was approved by the institutional ethical committee of the School of Management at Zhejiang University of Technology.

### Materials and procedure

The experiment procedure was shown in Fig 1a. At first, participants signed the informed consent, and then they received anodal tDCS or sham tDCS over the rVLPFC. The tDCS was delivered by using a constant current stimulator via saline-soaked surface sponge electrodes with a size of 5 x 5 cm. For anodal stimulation of the rVLPFC, the anode electrode was placed over F6 according to the international 10/20 EEG system [15, 23]. The cathodal electrode was placed above the contralateral supraorbital area (Fp1) approximately 5 cm from the anode [28, 29]. In the condition of anodal stimulation, a constant current of 2 mA (0.08mA/cm$^2$)

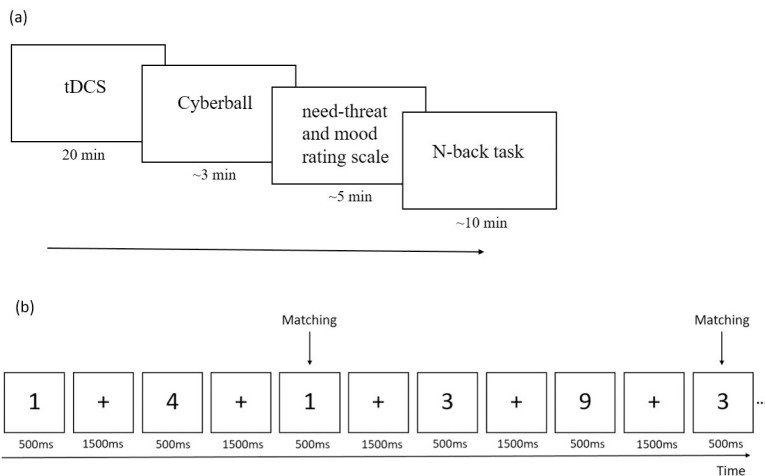

**Fig 1. The experiment procedure.** (a)Experiment procedure of the current study. Note that the Cyberball game was played at the last 5 minutes of the tDCS. (b)Example of the 2-back task.

intensity was applied for 20 minutes. According to previous studies, stimulation with 2 mA has been confirmed to be safe in healthy adult participants [15, 30, 31]. In the sham stimulation, the electrode was placed in the same position, but the stimulator was turned off automatically after 15 seconds and turned on for 15 seconds at the end of 20 minutes, totally lasting for 30 seconds. This mimicked the itching sensation of the anodal stimulation condition while providing negligible effects on neural activity. Thus, all participants believed they received stimulation for 20 minutes.

At the last 5 minutes of the anodal or sham stimulation, participants played the Cyberball game, which was widely used to manipulate social exclusion [32]. In this game, participants were told that they were playing with two other players (one male and one female). The degree of social exclusion or inclusion was manipulated by varying the number of times that participants received the ball from the other players. Specifically, in the social inclusion condition, participants received approximately one-third of the total throwed balls (i.e., 30 balls), while in the social exclusion condition, participants only received the ball twice at the beginning of the game. The Cyberball game lasted about 3 minutes.

Following the stimulation and the Cyberball game, participants completed a standard post-Cyberball questionnaire consisting of a need-threat scale and a mood rating schedulen [33]. For need-threat scale, participants were asked to assess their level of satisfaction for feelings of belonging, self-esteem, meaningful existence and control during the game on a five-point scale (1 = 'do not agree' to 5 = 'agree'). Lower scores represented an increased perceived threat to social need and indicated the effectiveness of the social exclusion manipulation. For the mood rating schedule, participants were asked to rate how good/bad, friendly/unfriendly, angry/pleasant, happy/sad they were currently feeling, on a scale from 1 (not at all) to 5 (very much). Negative items were re-coded. Based on previous studies [34, 35], we calculated an average mood rating score for each participant and each condition. Lower scores represented a decreased mood rating and suggested participants feel more negative emotion elicited by social exclusion.

Then, participants performed the N-back task, which was a common method to assess working memory [36, 37]. Similar with previous studies [10, 38], this study adopted a 2-back paradigm (see Fig 1b). Specifically, the numbers were flashed one-by-one on the screen for 500 ms with a 1500 ms interval. Participants were instructed to press the spacebar on the computer keyboard once the current number on the screen matched the number that appeared "two back" in the sequence. The conditions of matching or not were presented in a pseudorandom order with a 1:2 ratio. Participants completed 3 blocks of 40 trials each, and carried out a few practice trials before the formal task.

## Results

### Manipulation check

The need-threat scale scores were analyzed to check the effectiveness of the Cyberball manipulation. A 2 (Exclusionary Status: social inclusion vs social exclusion) × 2 (Stimulation Type: anodal vs sham stimulation) between-subject ANOVA revealed significant lower score for the exclusion group ($M_{exclusion}$ = 2.37, $SD_{exclusion}$ = 0.52) than that for the inclusion group ($M_{inclusion}$ = 3.99, $SD_{inclusion}$ = 0.44, $F(1, 124)$ = 356.29, $p < 0.001$, $\eta_p^2 = 0.74$). Besides, there was no significant difference between the anodal stimulation group ($M_{anodal}$ = 3.2, $SD_{anodal}$ = 0.96) and sham stimulation group ($M_{sham}$ = 3.17, $SD_{sham}$ = 0.93, $F(1, 124)$ = 0.13, $p = 0.72$). In addition, the interaction effect between Exclusionary Status and Stimulation Type was not significant ($F(1, 124)$ = 0.37, $p = 0.54$). These results suggested that the needs of excluded

participants were threatened compared to those of the included participants, confirming the effectiveness of the social exclusion manipulation.

## Mood ratings

A 2 (Exclusionary Status: social inclusion vs exclusion) × 2 (Stimulation Type: anodal vs sham stimulation) between-subject ANOVA showed that the main effect of Exclusionary Status was significant (F (1, 124) = 152.66, p < 0.001, $\eta_p^2 = 0.55$). The excluded participants had a lower mood rating score ($M_{exclusion}$ = 3.22, $SD_{exclusion}$ = 0.86) than included participants ($M_{inclusion}$ = 4.18, $SD_{exclusion}$ = 0.4). The main effect of Stimulation Type was also significant (F (1, 124) = 73.59, p < 0.001, $\eta_p^2 = 0.37$). Participants in the sham stimulation group had a lower mood rating score ($M_{sham}$ = 3.37, $SD_{sham}$ = 0.1) than participants in the anodal stimulation group ($M_{anodal}$ = 4.04, $SD_{anodal}$ = 0.38). In addition, there was a significant interaction effect between Exclusionary Status and Stimulation Type (F (1, 124) = 93.38, p < 0.001, $\eta_p^2 = 0.43$). As shown in Fig 2, the simple effect analyses showed that when receiving sham stimulation over the rVLPFC, social excluded participants had a significant lower mood rating score ($M_{exclusion\&sham}$ = 2.51, $SD_{exclusion\&sham}$ = 0.56) than included participants ($M_{inclusion\&sham}$ = 4.23, $SD_{inclusion\&sham}$ = 0.44, F (1, 124) = 242.42, p < 0.001, $\eta_p^2 = 0.66$). While receiving anodal stimulation over the rVLPFC, the difference in mood rating score between social excluded participants ($M_{exclusion\&anodal}$ = 3.93, $SD_{exclusion\&anodal}$ = 0.38) and included participants ($M_{inclusion\&anodal}$ = 4.14, $SD_{inclusion\&anodal}$ = 0.35) was marginally significant (F (1, 124) = 3.62, p = 0.06). The results indicated that under the anodal tDCS condition, the difference in mood rating score between excluded and included participants had a tendency to decrease. Besides, social excluded participants receiving anodal stimulation over the rVLPFC had a significant higher mood rating score ($M_{exclusion\&anodal}$ = 3.93, $SD_{exclusion\&anodal}$ = 0.38) than excluded participants given sham stimulation ($M_{exclusion\&sham}$ = 2.51, $SD_{exclusion\&sham}$ = 0.56, F (1, 124) = 166.39, p < 0.001, $\eta_p^2 = 0.57$). While for social included participants, no significant difference was observed between the anodal stimulation group ($M_{inclusion\&anodal}$ = 4.14, $SD_{inclusion\&anodal}$ = 0.35) and sham stimulation group ($M_{inclusion\&sham}$ = 4.23, $SD_{inclusion\&sham}$ = 0.44, F (1, 124) = 0.59, p = 0.44).

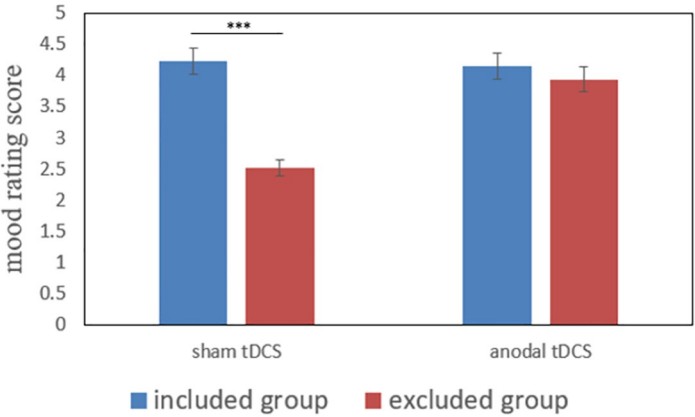

**Fig 2. Mood rating scores for excluded and included participants receiving anodal or sham stimulation.** Error bars represent SEMs. ***p < 0.001.

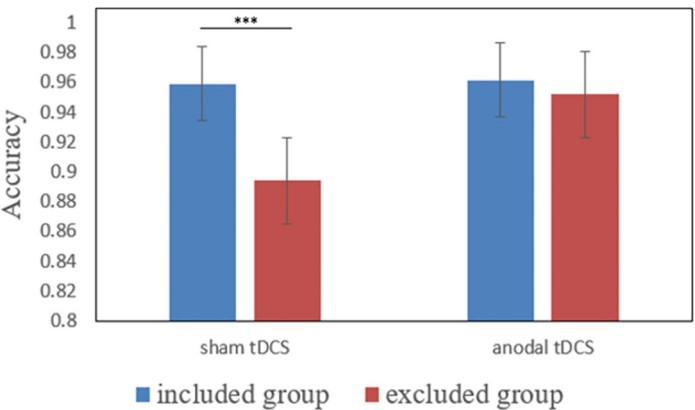

**Fig 3. The accuracy of 2-back task for excluded and included participants receiving anodal or sham stimulation.** Error bars represent SEMs. ***p < 0.001.

## WM performance

As previous studies, we mainly focused on the task accuracy to measure the WM performance [39, 40]. A 2 (Exclusionary Status: social inclusion vs exclusion) × 2 (Stimulation Type: anodal vs sham stimulation) between-subject ANOVA showed that the main effect of Exclusionary Status was significant (F (1, 124) = 7.83, p = 0.01, $\eta_p^2 = 0.06$). The included participants ($M_{inclusion}$ = 96.03%, $SD_{inclusion}$ = 0.06) exhibited a higher accuracy than excluded participants ($M_{exclusion}$ = 92.31%, $SD_{exclusion}$ = 0.09). The main effect of Stimulation Type was also significant (F (1,124) = 5.09, p = 0.03, $\eta_p^2 = 0.04$). Participants in the anodal stimulation group ($M_{anodal}$ = 95.67%, $SD_{anodal}$ = 0.07) exhibited a higher accuracy than participants in the sham stimulation group ($M_{sham}$ = 92.67%, $SD_{sham}$ = 0.09). Besides, there was a significant interaction effect between Exclusionary Status and Stimulation Type (F (1,124) = 4.31, p = 0.04, $\eta_p^2 = 0.03$). As shown in Fig 3, the simple effect analyses showed that when receiving sham stimulation over the rVLPFC, social included participants ($M_{inclusion\&sham}$ = 95.91%, $SD_{inclusion\&sham}$ = 0.06) exhibited a higher accuracy than excluded participants ($M_{exclusion\&sham}$ = 89.42%, $SD_{exclusion\&sham}$ = 0.1, F (1, 124) = 11.87, p < 0.001, $\eta_p^2 = 0.09$). While receiving anodal stimulation over the rVLPFC, no significant difference was observed between social excluded participants ($M_{exclusion\&anodal}$ = 95.19%, $SD_{exclusion\&anodal}$ = 0.07) and included participants ($M_{inclusion\&anodal}$ = 96.15%, $SD_{inclusion\&anodal}$ = 0.06, F (1, 124) = 0.26, p = 0.61). Besides, social excluded participants receiving anodal stimulation over the rVLPFC ($M_{exclusion\&anodal}$ = 95.19%, $SD_{exclusion\&anodal}$ = 0.07) exhibited a significant higher accuracy than excluded participants receiving sham stimulation ($M_{exclusion\&sham}$ = 89.42%, $SD_{exclusion\&sham}$ = 0.1, F (1, 124) = 9.38, p = 0.003, $\eta_p^2 = 0.07$). While for social included participants, no significant difference was observed between the anodal ($M_{inclusion\&anodal}$ = 96.15%, $SD_{inclusion\&anodal}$ = 0.06) and sham stimulation condition ($M_{inclusion\&sham}$ = 95.91%, $SD_{inclusion\&sham}$ = 0.06, F (1, 124) = 0.02, p = 0.9).

## Relationships between social exclusion, negative emotion and WM Performance

The analyses reported in the two preceding sections showed that when receiving sham stimulation over the rVLPFC, social excluded participants exhibited lower accuracy and lower mood

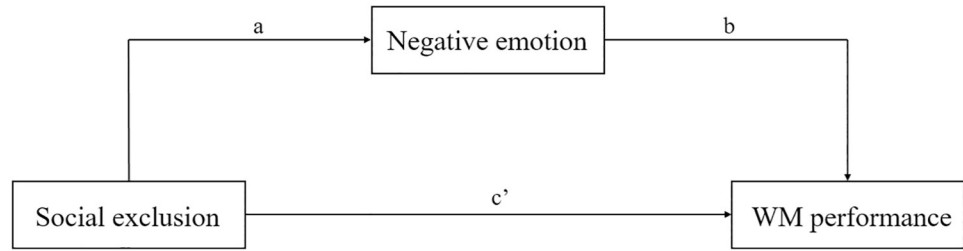

**Fig 4. Mediation model of negative emotion between social exclusion and WM performance.**

rating score than social included participants. To further probe the relationship between social exclusion, emotion and working memory, we applied a mediation model to analyze the data from sham stimulation group through the PROCESS SPSS computational tool [41] on Exclusionary Status (social exclusion or social included; independent variable [IV]), WM performance (accuracy in the N-back task; dependent variable [DV]), and emotion (mood rating scores; mediator [M]; see Fig 4). Given the limited sample size and to prevent the effects of violation of normal distribution assumptions, we used the nonparametric bootstrapping method as a robust estimation of both direct and indirect effects [42], which provided a confidence interval (CI) around the indirect effect of the IV on the DV via M. Moreover, in order to enhance confidence in our results, we used the bootstrapping method to obtain a sampling distribution for each parameter based on 5000 sample replications. Mediation is significant if the interval between the upper limit and lower limit of a bootstrapped 95% CI does not contain zero, which means that the mediating effect is significantly different from zero [41].

As shown in Table 1, the results indicated that emotion fully mediated the link between Exclusionary Status and WM performance. In other words, our results showed that social exclusion induces negative emotion, which in turn lead to impaired WM performance. To test the alleviating effect of anodal stimulation over the rVLPFC on such adverse effect of social exclusion, we further analyzed the mediation model with the data from the anodal stimulation condition. As shown in Table 2, no mediating effect was found.

## Discussion

The aim of this study was to explore the emotional mechanism underlying the adverse effect of social exclusion on working memory (WM) performance, and test whether anodal tDCS over the rVLPFC could alleviate this adverse effect. The participants completed the 2-back task in the conditions of exclusionary status (social inclusion vs. social exclusion) × stimulation (anodal vs. sham stimulation over the rVLPFC). ANOVA results showed that under the sham tDCS condition, mood rating score and 2-back task accuracy of social excluded participants were both lower than those of social included participants. Besides, mood rating score and

**Table 1. Mediation analysis of emotion between social exclusion and WM performance in the sham stimulation condition.**

| Parameter | Point estimate (SE) | 95% CI |
|---|---|---|
| Direct effect (c') | 0.086 (0.036) | [-0.015, 0.158] |
| Indirect effect (a × b) | -0.151 (0.035) | [-0.221, -0.084] |

Note. CI = Confidence Interval. If the CI includes zero, then the point estimate is not statistically significant.

**Table 2. Mediation analysis of emotion between social exclusion and WM performance in the anodal stimulation condition.**

| Parameter | Point estimate (SE) | 95% CI |
| --- | --- | --- |
| Direct effect (c') | -0.003 (0.017) | [-0.037, 0.031] |
| Indirect effect (a × b) | -0.007 (0.006) | [-0.021, 0.003] |

Note. CI = Confidence Interval. If the CI includes zero, then the point estimate is not statistically significant.

2-back task accuracy of excluded participants receiving anodal tDCS over the rVLPFC were both higher than excluded participants receiving sham tDCS. Furthermore, the mediation model indicated that negative emotion mediated the relationship between social exclusion and WM performance under the sham tDCS condition, while this mediating effect disappeared under the anodal tDCS condition. These findings supported our hypotheses that (1) the adverse effect of social exclusion on WM performance was mediated by negative emotion, and (2) anodal tDCS over the rVLPFC could successfully alleviate the adverse effect by reducing negative emotion.

Previous studies suggested that resource-consuming self-regulatory process such as emotion regulation might be a candidate mechanism for the adverse effect of social exclusion on WM performance [16, 42]. However, there are less empirical evidences that directly support this assumption. Some studies discovered that depressed individuals also exhibited the impaired WM performance when they faced social exclusion [43–45]. As we know, dysregulation of emotion is one of the core features of depression [46], and it might imply that it is the emotion per se rather than emotion regulation that leads to impaired WM performance. Therefore, we put forward the emotion explanation and our findings confirmed that the negative emotion could account for the adverse effect of social exclusion on WM performance, which provides contribution to the understanding of the relationships between social exclusion, emotion and WM performance. Besides, our study adopted verbal material (i.e. numbers) as stimuli and suggested that negative emotions elicited by social exclusion would impair verbal WM performance, which extends the existing studies regarding the relationships between negative emotions and different subsets of WM. Some studies pointed out that there was a selective effect of negative emotion on visual-spatial and verbal WM performance, and suggested that negative emotion had a greater impact on visual-spatial WM than verbal WM [47]. These studies usually used pictures of blood and violence or electric shock threats as stimuli to induce negative emotions, mostly manifested as fear and anxiety. According to neuroimaging studies, anxiety would induce greater activity in the right parietal lobe [48] and the right prefrontal cortex [49], and fear would lead to rapid activation of activity in the right amygdala [50]. Besides, visual-spatial WM is mainly involved in the right hemisphere, while verbal WM is mainly involved in the left hemisphere [51]. Therefore, it might be the overlap of right brain regions that leads to the greater impact of negative emotions on visual-spatial WM in previous studies. By contrast, in our study, the negative emotions induced by the Cyberball paradigm were more complex, including angry, sadness, sham and so on [52, 53]. A fMRI study conducted by Peterson et al. [54] adopted Cyberball paradigm to manipulate social exclusion, and revealed that the angry responses induced by social exclusion would lead to more activity in the left prefrontal cortex. Besides, a scoping review conducted by Wang et al. [55] concluded that the bilateral prefrontal cortex was activated during social exclusion. Considering that verbal WM performance relies on the activity in the left prefrontal cortex [51, 56], the overlap of activation in left brain regions might lead to the adverse effect of social exclusion on verbal WM.

Although previous studies have found the bilateral VLPFC activity increased in response to social exclusion [15, 20, 55], some studies applied the transcranial magnetic stimulation (TMS) on the left PFC and reported no effect on self-reported negative emotion [57], while other studies showed that anodal tDCS over the rVLPFC mitigated negative emotion during social exclusion [22]. The results of the present study showed that social excluded participants receiving sham stimulation over the rVLPFC had lower emotion rating score than excluded participants receiving anodal stimulation, which confirms the causal relationship between the rVLPFC activity and self-reported negative emotion [23]. Furthermore, our results also showed that mood rating score and WM task accuracy of excluded participants receiving anodal tDCS over the rVLPFC were both significantly higher than excluded participants receiving sham tDCS, and the mediating role of negative emotion disappeared, demonstrating the reduced adverse effect of social exclusion attributed to the anodal stimulation over the rVLPFC. The adverse effect of social exclusion on individuals is mainly reflected in three aspects: emotion [3], behavior [58] and cognition [9]. In addition to the negative emotion, previous studies also have proved anodal tDCS over the rVLPFC could reduce the adverse behavioral effects of social exclusion, such as a decrease in aggressive behavior [58]. The current study complementally revealed that anodal tDCS over the rVLPFC could also reduce the adverse effect of social exclusion on the WM performance.

Although our study provided primary insights into the mediating role of negative emotion in the adverse effect of social exclusion on WM performance and anodal tDCS over the rVLPFC could reduce this adverse effect, it is also restricted by several limitations that could be remedied in the future work. First, according to the Multicomponent Model proposed by Baddeley [11], WM can be divided into verbal WM and visual-spatial WM. Our study adopted verbal material (i.e. numbers) as stimuli and only assessed verbal WM, which may limit the extension of the conclusions about the effect of social exclusion to the visual-spatial WM. Second, our study didn't consider other brain regions involved in social exclusion and emotion regulation. For example, in the context of social exclusion, two brain regions, including the ventral anterior cingulate cortex (VACC) and rVLPFC, have been implicated in regulation of negative emotion [15]. Third, we only delivered anodal tDCS and ignored cathodal tDCS. In order to further understand the role of rVLPFC in the impairment of social exclusion on WM performance, we will take cathodal tDCS into account to test the inhibitory aspect in our future research. Besides, we suggest that future studies could adopt visual-spatial material (e.g., color squares) as stimuli to further explore whether negative emotion also mediates the relationship between social exclusion and visual-spatial WM performance, and examine whether anodal tDCS over other brain regions (e.g., VACC) could also reduce the adverse effect of social exclusion on WM performance.

## Conclusion

In summary, the current study investigated the relationships between social exclusion, emotion and WM performance, and confirmed a mediation model in which negative emotion elicited by social exclusion mediates the adverse effect of social exclusion on WM performance. Furthermore, this study, to our knowledge, firstly verified that anodal stimulation over the rVLPFC could reduce such adverse effect. This study could help to extend the theoretical research on the influence of social exclusion on individuals and provide a clinical treatment in response to social exclusion.

## Supporting information

**S1 Dataset. Experimental data.**
(XLSX)

## Author Contributions

**Conceptualization:** Qingguo Ma.

**Data curation:** Yu Pang.

**Formal analysis:** Yu Pang.

**Funding acquisition:** Qingguo Ma.

**Methodology:** Yu Pang.

**Software:** Yu Pang.

**Supervision:** Qingguo Ma.

**Validation:** Qingguo Ma.

**Visualization:** Yu Pang.

**Writing – original draft:** Yu Pang.

**Writing – review & editing:** Qingguo Ma.

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
