## [Decision Letter · Decision Letter 0]

27 Jan 2023

PONE-D-22-35184The Emotional Mechanism Underlying the Adverse Effect of Social Exclusion on Working Memory Performance: a tDCS studyPLOS ONE

Dear Dr. Ma,

Thank you for submitting your manuscript to PLOS ONE. After careful consideration, we feel that it has merit but does not fully meet PLOS ONE’s publication criteria as it currently stands. Therefore, we invite you to submit a revised version of the manuscript that addresses the points raised during the review process.

Please submit your revised manuscript by Mar 13 2023 11:59PM. If you will need more time than this to complete your revisions, please reply to this message or contact the journal office at plosone@plos.org. Please include the following items when submitting your revised manuscript:A rebuttal letter that responds to each point raised by the academic editor and reviewer(s). You should upload this letter as a separate file labeled 'Response to Reviewers'.A marked-up copy of your manuscript that highlights changes made to the original version. You should upload this as a separate file labeled 'Revised Manuscript with Track Changes'.An unmarked version of your revised paper without tracked changes. You should upload this as a separate file labeled 'Manuscript'.If applicable, we recommend that you deposit your laboratory protocols in protocols.io to enhance the reproducibility of your results. Protocols.io assigns your protocol its own identifier (DOI) so that it can be cited independently in the future. For instructions see: https://journals.plos.org/plosone/s/submission-guidelines#loc-laboratory-protocols. Additionally, PLOS ONE offers an option for publishing peer-reviewed Lab Protocol articles, which describe protocols hosted on protocols.io. Read more information on sharing protocols at https://plos.org/protocols?utm_medium=editorial-email&utm_source=authorletters&utm_campaign=protocols.

We look forward to receiving your revised manuscript.

Kind regards,

Irene Cristofori

Academic Editor

PLOS ONE

Journal Requirements:

"This study was supported by the National Natural Science Foundation of China (No.71942004 and 72002202), and the Humanities and Social Sciences Foundation of the Ministry of Education of China (No. 20YJC630040)"

"This study was supported by the National Natural Science Foundation of China (No.71942004 and 72002202), and the Humanities and Social Sciences Foundation of the Ministry of Education of China (No. 20YJC630040). The funders had no role in study design, data collection and analysis, decision to publish, or preparation of the manuscript."

Reviewers' comments:

Reviewer's Responses to Questions

**Comments to the Author**

1. Is the manuscript technically sound, and do the data support the conclusions?

Reviewer #1: Partly

2. Has the statistical analysis been performed appropriately and rigorously? 

Reviewer #1: Yes

3. Have the authors made all data underlying the findings in their manuscript fully available?

Reviewer #1: Yes

4. Is the manuscript presented in an intelligible fashion and written in standard English?

Reviewer #1: Yes

5. Review Comments to the Author

Reviewer #1: In this manuscript, Qingguo and Yu investigated whether social exclusion has an adverse effect on WM performance, and whether this effect is mediated by the negative emotion induced by the social exclusion. Before participants performed a Cyberball task, they received sham or anodal tDCS over the rVLPFC. Immediately after that, participants fulfil a need-threat scale and mood rating questionnaire, to finally perform a 2-back task.

As authors suggested, after sham stimulation, participants in an exclusion condition in the Cyberball task, presented lower scores in the mood rating questionnaire and the performance in the WM task was worse compared to included participants. Suggesting that negative emotion modulates the worsening in WM task. On the other hand, anodal tDCS over rVLPFC abolished such effect.

This study addresses an interesting question i.e., whether the adverse effect on WM provoked by social exclusion, is modulated by negative emotion exclusion-induced.

The manuscript is overall well written, procedures are well designed and overall methods sound. In sum, I think this manuscript is appropriate PLOSE ONE journal.

I have a few minor comments that I think authors should consider in a revision.

How did the authors determine the size sample? Did they use Gpower, did they refer to

previous literature? I think they should add a short statement clarifying this aspect.

There is an aspect that a bit crucial to me: the difference in mood rating between excluded and included participants, after receiving anodal stimulation was marginally significant (p = 0.06). I consider that the authors should discuss this issue, since I don’t think that they can strongly claim that the anodal tDCS reduced the negative emotion induced by social exclusion compared to social inclusion. This result, consequently affects the conclusion that the performance in WM is mainly modulate by low mood rating. In fact, the main conclusion that the authors can state is that anodal tDCS reverse the effect of social exclusion on the WM, but they cannot conclude that such modulation is principally due to the mood rating.

A curiosity: why the authors only considered to test the excitatory effect of the stimulation, and they didn’t take into account the inhibitory aspect, delivering cathodal tDCS over the rVLPFC?

Please proof read the text: in some cases, formulations could be improved.

6. PLOS authors have the option to publish the peer review history of their article (what does this mean?). If published, this will include your full peer review and any attached files.

Reviewer #1: No

---

## [Author Response · Author response to Decision Letter 0]

22 Mar 2023

Dear editor and reviewer,

We appreciate the valuable comments and suggestions given by the reviewer, which have been conscientiously considered in our amendment of the manuscript and significantly improve the manuscript. All the changes made to the manuscript are marked with Track Changes in the revised manuscript. Our detailed responses to the comments of the reviewers are given below.

Response to reviewer #1

Point 1. How did the authors determine the size sample? Did they use Gpower, did they refer to previous literature? I think they should add a short statement clarifying this aspect.

Response: Thanks for reviewer’s suggestions. The way we determined the size sample was according to the calculation of G*Power 3.1 (Faul et al., 2009). Under the premise of power (1-β) =0.95, α = 0.05 and effect size f = 0.4 (Weintraub-Brevda & Chua, 2019), a minimum of 84 participants were needed, with 21 in each group. As a reference, Riva et al. (2015) recruited 80 participants with 20 in each group and He et al. (2018) recruited 44 participants with 22 in each group. Based on this, 128 participants were recruited in our research with 32 in each group. A short statement clarifying how we determined the size sample is added in the Participants section (page 3) in the 'Revised Manuscript with Track Changes' file. 

Reference: 

[1] Faul, F., Erdfelder, E., Buchner, A., & Lang, A. G. Statistical power analyses using G* Power 3.1: Tests for correlation and regression analyses. Behavior research methods. 2009;41(4):1149-1160.

[2] Weintraub-Brevda, R. R., & Chua, E. F. Transcranial direct current stimulation over the right and left VLPFC leads to differential effects on working and episodic memory. Brain and Cognition. 2019;132:98-107.

Point 2. There is an aspect that a bit crucial to me: the difference in mood rating between excluded and included participants, after receiving anodal stimulation was marginally significant (p = 0.06). I consider that the authors should discuss this issue, since I don’t think that they can strongly claim that the anodal tDCS reduced the negative emotion induced by social exclusion compared to social inclusion. This result, consequently affects the conclusion that the performance in WM is mainly modulate by low mood rating. In fact, the main conclusion that the authors can state is that anodal tDCS reverse the effect of social exclusion on the WM, but they cannot conclude that such modulation is principally due to the mood rating.

Response: Thanks for reviewer’s suggestions. We agree with your opinion that we can’t strongly claim that the anodal tDCS reduced the negative emotion induced by social exclusion compared to social inclusion based on the marginally significant result (p = 0.06). It might indicate that the difference in mood rating score between excluded and included participants had a tendency to decrease under anodal tDCS. We have modified the description in the Result section (page 5) in the 'Revised Manuscript with Track Changes' file. 

In order to probe the relationship between social exclusion, negative emotion and WM performance, we applied a mediation model to analyze the data from sham stimulation group through the PROCESS SPSS computational tool. The results indicated that the impairment of social exclusion on WM performance was fully mediated by negative emotion. Besides, we applied a 2 × 2 between-subject ANOVA to test the alleviating effect of anodal tDCS over the rVLPFC on the adverse effect of social exclusion on WM performance. The ANOVA results showed that mood rating score and 2-back task accuracy of excluded participants receiving anodal tDCS over the rVLPFC were both significantly higher than excluded participants receiving sham tDCS. Besides, we also analyzed the mediation model with the data from the anodal stimulation condition and found no more mediating effect. These results contributed to our second conclusion that anodal tDCS over the rVLPFC would alleviate the adverse effect of social exclusion on WM performance by reducing the negative emotion.

Point 3. A curiosity: why the authors only considered to test the excitatory effect of the stimulation, and they didn’t take into account the inhibitory aspect, delivering cathodal tDCS over the rVLPFC? 

Response: Thanks for reviewer’s suggestions. One of the aims of this study was to investigate the potential solution of neural intervention to reduce the impairment of social exclusion on working memory performance. Based on the findings from previous studies that (1) social exclusion would induce negative emotion (e.g., anxiety, sadness, depression), and (2) negative emotion would impair WM performance, we first hypothesized that negative emotions induced by social exclusion mediate the relationship between social exclusion and WM performance. Besides, according to the findings from previous studies that anodal stimulation over the rVLPFC would reduce the negative emotion, we further hypothesized that anodal stimulation over the rVLPFC would alleviate the adverse effect of social exclusion on WM performance. Along this line, we only considered to test the excitatory effect of the stimulation because it was in line with our study purpose i.e. investigate the potential solution of neural intervention to alleviate the impairment of social exclusion on WM performance. 

According to reviewer’s suggestion and for further understanding the role of rVLPFC in the impairment of social exclusion on WM performance, we will take cathodal tDCS into account to test the inhibitory aspect in our future research. We have added this future research direction in the Discussion section (page 9) in the 'Revised Manuscript with Track Changes' file. 

Sincerely,

Qingguo Ma on behalf of the co-authors.

---

## [Editor Report · Decision Letter 1]

28 Mar 2023

The Emotional Mechanism Underlying the Adverse Effect of Social Exclusion on Working Memory Performance: a tDCS study

PONE-D-22-35184R1

Dear Dr. Ma,

We’re pleased to inform you that your manuscript has been judged scientifically suitable for publication and will be formally accepted for publication once it meets all outstanding technical requirements.

Kind regards,

Irene Cristofori

Academic Editor

PLOS ONE
---

## [Editor Report · Acceptance letter]

31 Mar 2023

PONE-D-22-35184R1 

The Emotional Mechanism Underlying the Adverse Effect of Social Exclusion on Working Memory Performance: a tDCS study 

Dear Dr. Ma:

I'm pleased to inform you that your manuscript has been deemed suitable for publication in PLOS ONE. Congratulations! Your manuscript is now with our production department. 

Kind regards, 

on behalf of

Dr. Irene Cristofori 

Academic Editor

PLOS ONE